# Discrete JEPA: Learning Discrete Token Representations without Reconstruction

**Junyeob Baek** [1]  **Hosung Lee** [1]  **Christopher Hoang** [2]  **Mengye Ren** [2]  **Sungjin Ahn** [1,2]

## Abstract

The cornerstone of cognitive intelligence lies in extracting hidden patterns from observations and leveraging these principles to systematically predict future outcomes. However, current image tokenization methods demonstrate significant limitations in tasks requiring symbolic abstraction and logical reasoning capabilities essential for systematic inference. To address this challenge, we propose Discrete-JEPA, extending the latent predictive coding framework with semantic tokenization and novel complementary objectives to create robust tokenization for symbolic reasoning tasks. Discrete-JEPA dramatically outperforms baselines on visual symbolic prediction tasks, while striking visual evidence reveals the spontaneous emergence of deliberate systematic patterns within the learned semantic token space. Though an initial model, our approach promises a significant impact for advancing Symbolic world modeling and planning capabilities in artificial intelligence systems.

## 1. Introduction

The ability to extract meaningful patterns from visual observations and systematically predict future outcomes represents a cornerstone of cognitive intelligence, forming the foundation for what cognitive scientists term System 2 reasoning—deliberate, systematic thinking that enables complex planning and problem-solving (Kahneman, 2011; Evans & Stanovich, 2013; Bengio et al., 2019). In artificial intelligence, this capability translates to the fundamental challenge of developing world models that can perform symbolic abstraction and logical reasoning (Goyal et al., 2021; Sehgal et al., 2023; Tang et al., 2024; Baek et al., 2025), enabling agents to plan effectively over extended temporal horizons.

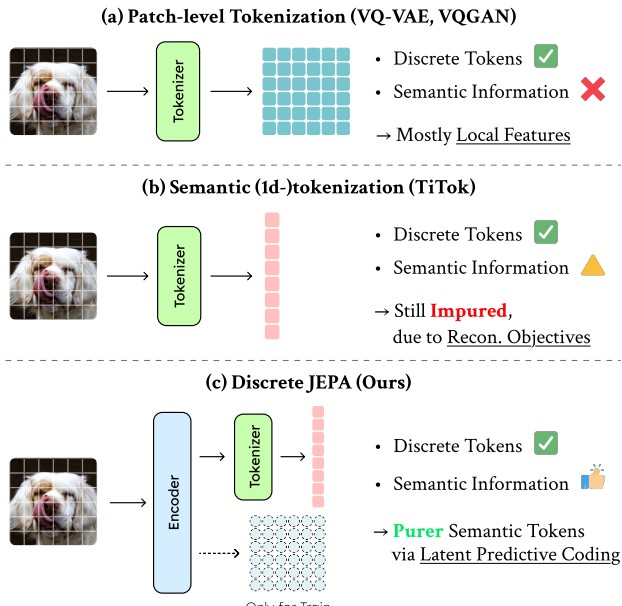

*Figure 1.* **Discrete JEPA Overview.** Existing tokenization approaches suffer from limited semantic abstraction **(a)** or reconstruction bias **(b)**. Our Discrete JEPA addresses both limitations by learning discrete semantic tokens via latent predictive coding, enabling superior symbolic reasoning capabilities.

Recent advances in image tokenization (Van Den Oord et al., 2017; Esser et al., 2021; Ramesh et al., 2021; Razavi et al., 2019; Yu et al., 2021) and autoregressive modeling (Esser et al., 2021; Chang et al., 2022; Yu et al., 2023; Yan et al., 2023) have demonstrated remarkable progress in visual understanding and generation tasks. However, these approaches primarily focus on patch-level local feature tokenization, which, while effective for reconstruction and generation, exhibit significant limitations when applied to tasks requiring symbolic reasoning and logical planning capabilities. The granular nature of patch-based representations introduces computational overhead and, more critically, fails to capture the high-level semantic abstractions necessary for systematic inference and long-horizon planning.

Contemporary efforts to address these limitations have explored semantic-level tokenization approaches, such as Yu et al. (2024); Wu et al. (2024); Kim et al. (2025); Bachmann et al. (2025), which attempt to move beyond patch-level representations toward more meaningful 1D tokenization.

[1]KAIST [2]New York University. Correspondence to: Junyeob Baek <wnsdlqjtm@kaist.ac.kr>, Sungjin Ahn <sungjin.ahn@kaist.ac.kr>.

*Proceedings of the ICML 2025 Tokenization Workshop (TokShop)*, Vancouver, Canada. PMLR 267, 2025. Copyright 2025 by the author(s).

However, these methods remain constrained by their reliance on pixel-level reconstruction objectives, resulting in tokens that encode unnecessary visual details rather than the abstract semantic concepts crucial for symbolic reasoning. This fundamental mismatch between the granularity of representation and the requirements of symbolic planning tasks represents a significant barrier to developing truly intelligent visual reasoning systems.

The Joint-Embedding Predictive Architecture (JEPA) framework (LeCun, 2022; Assran et al., 2023; Bardes et al., 2023b; Sobal et al., 2022) offers a promising alternative by learning representations through latent-space prediction rather than pixel-level reconstruction. By predicting masked representations in latent space, Assran et al. (2023) demonstrates the potential for learning more semantically meaningful features. However, the continuous nature of its representations limits their applicability to autoregressive modeling paradigms, where discrete tokens are essential for effective sequence modeling and long-horizon prediction with reduced accumulated error.

To bridge this gap, we propose *Discrete-JEPA*, a novel extension of the JEPA framework that learns discrete semantic tokens capturing high-level semantic abstractions while preserving the benefits of latent-space predictive learning. Our approach introduces semantic-level vector quantization to the JEPA architecture while maintaining the framework's core advantage of latent-space predictive learning. Through a carefully designed unified predictive framework, Discrete-JEPA learns to encode global semantic information into discrete tokens while preserving fine-grained spatial details through complementary continuous representations.

Our contributions are threefold: (1) We introduce the Discrete-JEPA architecture, which extends the JEPA framework with semantic tokenization and novel complementary objectives (Semantic-to-Patch, Patch-to-Semantic, and Patch-to-Patch prediction) to learn robust discrete semantic tokens for enhanced representation learning. (2) We demonstrate that Discrete-JEPA significantly outperforms existing baselines across challenging visual symbolic prediction tasks, validating the effectiveness of our semantic tokenization approach. (3) We provide compelling visual evidence of systematic patterns that emerge within the learned semantic token space, offering insights into the model's representation capabilities and potential for more complex reasoning tasks.

## 2. Related Works

**Self-supervised Visual Representation Learning.** Self-supervised learning has evolved through contrastive learning (Chen et al., 2020; He et al., 2020; Caron et al., 2020), variance-based regularization (Bardes et al., 2021), boot-strap methods (Grill et al., 2020), self-distillation (Caron et al., 2021; Oquab et al., 2024), and masked reconstruction approaches (He et al., 2022; Bao et al., 2021; Zhou et al., 2022). Recent work has also explored unified multimodal frameworks (Baevski et al., 2022). While these methods achieve strong performance on recognition tasks, they predominantly learn patch-level embeddings optimized for local features rather than the global semantic abstractions required for symbolic reasoning. Our approach addresses this limitation by learning semantic-level discrete tokens that capture high-level conceptual information.

**Discrete Image Tokenization.** Discrete visual representations emerged with VQ-VAE (Van Den Oord et al., 2017) and subsequent vector quantization methods (Esser et al., 2021; Yu et al., 2021), enabling token-based autoregressive generation (Ramesh et al., 2021; Chang et al., 2022). Building upon these foundations, researchers have developed alternative quantization schemes (Lee et al., 2022; Van Balen & Levy, 2019; Takida et al., 2022; Mentzer et al., 2023) and extended tokenization to video domains (Yu et al., 2023). More recently, semantic-level approaches have explored 1D tokenization (Yu et al., 2024; Chen et al., 2025b; Wang et al., 2025; Bachmann et al., 2025). However, reliance on pixel-level reconstruction objectives biases representations toward fine-grained details rather than semantic concepts essential for symbolic reasoning. We overcome this limitation through latent predictive learning that avoids reconstruction bias.

**Joint-Embedding Predictive Architectures.** JEPA (LeCun, 2022) introduced latent-space prediction as an alternative to pixel reconstruction. I-JEPA (Assran et al., 2023) demonstrated superior sample efficiency through masked representation prediction, inspiring extensions to audio (Fei et al., 2023), video (Bardes et al., 2023a), multi-modal motion-content learning (Bardes et al., 2023b), and diffusion applications (Chen et al., 2025a). Despite these advances, continuous representations suffer from accumulated errors in sequential prediction and lack discrete structure necessary for robust symbolic reasoning. Our work extends JEPA with discrete semantic tokenization and complementary predictive objectives to enable stable long-horizon prediction.

## 3. Preliminaries

**Joint-Embedding Predictive Architecture.** Joint-Embedding Predictive Architecture (JEPA) (LeCun, 2022; Assran et al., 2023) learns representations by predicting masked portions of the input in representation space rather than pixel space. Specifically, Assran et al. (2023) employs three key components: a context encoder $f_\theta^c$, a target encoder $f_{\bar{\theta}}^t$, and a predictor $g_\phi$.

Given an input image $x \in \mathbb{R}^{H \times W \times C}$, the image is divided

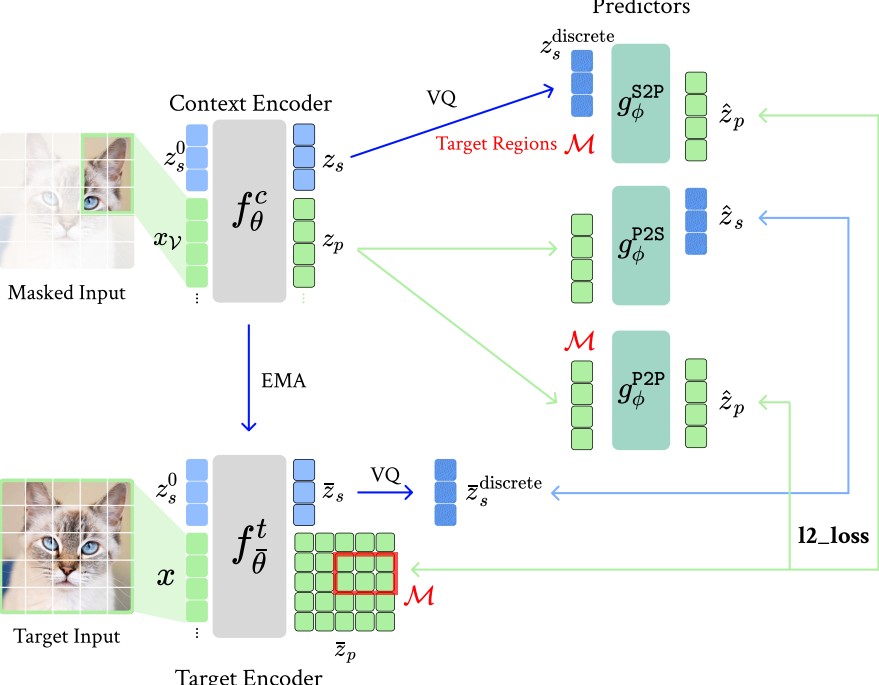

*Figure 2.* **Discrete-JEPA Architecture Overview.** The context encoder $f_\theta^c$ takes masked inputs with learnable tokens $z_s^0$ and generates semantic ($z_s$) and patch ($z_p$) representations, while the target encoder $f_{\bar\theta}^t$ processes the complete image to produce target representations $\bar{z}_s$ and $\bar{z}_p$. Vector quantization (VQ) is applied only to semantic representations to create discrete tokens $z_s^{\text{discrete}}$. Using these discrete semantic tokens and continuous patch tokens, the model performs three complementary prediction tasks (S2P, P2S, P2P) and compares predictions against the target encoder outputs, whose parameters are updated via EMA.

into patches and processed as follows:

1.  **Context Processing**: Visible patches (context block) $x_\mathcal{V}$ are encoded by the context encoder to obtain context representations $z_c = f_\theta^c(x_\mathcal{V})$.

2.  **Target Processing**: The entire image is processed by the target encoder to obtain actual patch representations at target locations $z_t = f_{\bar\theta}^t(x_\mathcal{M})$.

3.  **Prediction**: The predictor takes context representations and target position indices to predict what representations should exist at those target locations: $\hat{z}_t = g_\phi(z_c, \mathcal{M})$, where $\mathcal{M}$ contains the positional indices of target patches.

The training objective minimizes the L2 distance between predicted and target representations:

$$\mathcal{L}_{\text{I-JEPA}} = \sum_{i\in\mathcal{M}} ||f_{\bar\theta}^t(x_i) - g_\phi(f_\theta^c(x_\mathcal{V}), i)||_2^2 \quad (1)$$

where $i$ represents the positional index of target patches, and the predictor $g_\phi$ takes both the context representations and the target position index $i$ to predict what should be at that location. The target encoder $f_{\bar\theta}^t$ processes the actual patches

to provide the ground truth representations for comparison. Crucially, the target encoder parameters are updated via exponential moving average (EMA) of the context encoder, as shown in (He et al., 2020; Caron et al., 2021).

## 4. Discrete JEPA Tokenization

We propose Discrete JEPA, which extends the Joint-Embedding Predictive Architecture to learn discrete semantic tokens for symbolic reasoning and long-horizon planning. Our approach discretizes only semantic representations while maintaining continuous patch representations as intermediate features during training.

The method comprises three key components: an extended JEPA framework (Section 4.1), a semantic and patch tokenization strategy (Section 4.2), and complementary predictive objectives (Section 4.3).

### 4.1. Architecture

Our approach builds upon the JEPA framework (Assran et al., 2023), which employs three key components: a context encoder $f_\theta^c$, a target encoder $f_{\bar\theta}^t$, and predictors $g_\phi$. We extend this architecture to support *semantic-level discrete tokenization* while preserving the original spatial prediction

capabilities.

Given an input image $x \in \mathbb{R}^{H \times W \times C}$, our Discrete JEPA processes the image with the following components:

**Context Encoder** $f_\theta^c$: Processes visible image patches $x_\mathcal{V}$, sampled from patched inputs $\{x_i\}_{i=0}^{N_p}$ according to masking strategies, to obtain semantic and patch-level representations $z_s, z_p$:

$$z_s, z_p = f_\theta^c(z_s^0, x_\mathcal{V}) \tag{2}$$

where $z_s^0$ consists of $L$ learnable tokens.

**Target Encoder** $f_{\bar{\theta}}^t$: Processes the entire image $x$ along with learnable tokens $z_s^0$ to generate target semantic and patch representations $\bar{z}_s, \bar{z}_p$:

$$\bar{z}_s, \bar{z}_p = f_{\bar{\theta}}^t(z_s^0, x) \tag{3}$$

**Vector Quantization**: Applies vector quantization to semantic representations from both encoders to obtain discrete semantic tokens using a shared semantic codebook $\mathcal{C}_s \in \mathbb{R}^{K_s \times D_s}$:

$$z_s^{\text{discrete}} = \text{VQ}(z_s), \quad \bar{z}_s^{\text{discrete}} = \text{VQ}(\bar{z}_s) \tag{4}$$

**Predictors** $g_\phi$: Process semantic and patch tokens $z_s^{\text{discrete}}, z_p$ with target masks $\mathcal{M}$ to generate predictions for their respective objectives:

$$\hat{z}_p = g_\phi^{\text{S2P}}(z_s^{\text{discrete}}, \mathcal{M}), \hat{z}_s = g_\phi^{\text{P2S}}(z_p), \hat{z}_p = g_\phi^{\text{P2P}}(z_p, \mathcal{M}) \tag{5}$$

Figure 2 illustrates the complete architecture and information flow of our Discrete JEPA framework.

### 4.2. Semantic and Patch Tokenization

Our approach employs two distinct types of tokens, each serving specific functional roles within the learning framework:

**Semantic Tokens (Discrete).** The semantic representation $\bar{z}_s$ captures global image context and is discretized through vector quantization to produce discrete semantic tokens. Given the continuous representation $\bar{z}_s \in \mathbb{R}^{D_s}$ and a learnable codebook $\mathcal{C}_s = c_1, c_2, ..., c_{K_s} \subset \mathbb{R}^{D_s}$ with $K_s$ prototypes, we find the nearest entry:

$$\bar{k}^* = \arg\min_{k \in 1,...,K_s} ||\bar{z}_s - c_k||_2 \tag{6}$$

The discrete semantic token is then:

$$\bar{z}_s^{\text{discrete}} = C(\bar{k}^*) = c_{\bar{k}^*}. \tag{7}$$

These discrete tokens serve as the primary output for downstream symbolic reasoning and long-horizon planning tasks.

For training, we follow standard vector quantization procedures with commitment loss and exponential moving average updates, following (Van Den Oord et al., 2017; Esser et al., 2021).

**Patch Tokens (Continuous).** We maintain continuous patch tokens $\bar{z}_p$ that capture fine-grained spatial details from the encoder $f_{\bar{\theta}}^t(z_s^0, x)$. Unlike discrete semantic tokens, patch tokens remain continuous and serve exclusively as intermediate representations during training. These continuous tokens facilitate effective information flow between semantic and spatial levels through our unified predictive framework, but are not used in the final tokenized output.

**Semantic-Patch Interaction.** The interaction between discrete semantic tokens ($\bar{z}_s^{\text{discrete}}$) and continuous patch tokens ($\bar{z}_p$) from encoders forms the foundation for our unified predictive training framework. Discrete semantic tokens provide global context that guides spatial prediction, while continuous patch tokens contribute local details that enhance semantic understanding. This bidirectional relationship enables effective learning between global and local representations, setting the stage for the complementary predictive objectives detailed in the following section.

### 4.3. Complementary Predictive Objectives

We introduce three predictive objectives that operate between discrete semantic tokens and continuous patch tokens, each serving a distinct role in learning meaningful discrete semantic tokens:

**Semantic-to-Patch (S2P) Prediction.** The S2P objective encourages discrete semantic tokens to encode sufficient global context by predicting continuous patch tokens at target locations:

$$\mathcal{L}_{\text{S2P}} = \sum_{i \in \mathcal{M}} ||\bar{z}_p^{(i)} - g_\phi^{\text{S2P}}(z_s^{\text{discrete}}, i)||_2^2. \tag{8}$$

where $z_s^{\text{discrete}}$ is the discrete semantic token and $i$ encodes the spatial position. This objective enables the model to learn how global semantic information relates to local spatial details.

**Patch-to-Semantic (P2S) Prediction.** The P2S objective learns to extract semantic abstractions from continuous patch tokens:

$$\mathcal{L}_{\text{P2S}} = ||\bar{z}_s - g_\phi^{\text{P2S}}(z_p)||_2^2. \tag{9}$$

This objective encourages continuous patch tokens to contribute meaningfully to global semantic understanding, ensuring consistency between continuous and discrete token representations.

**Patch-to-Patch (P2P) Prediction.** The P2P objective maintains spatial coherence by predicting continuous patch tokens from other continuous patch tokens, following the

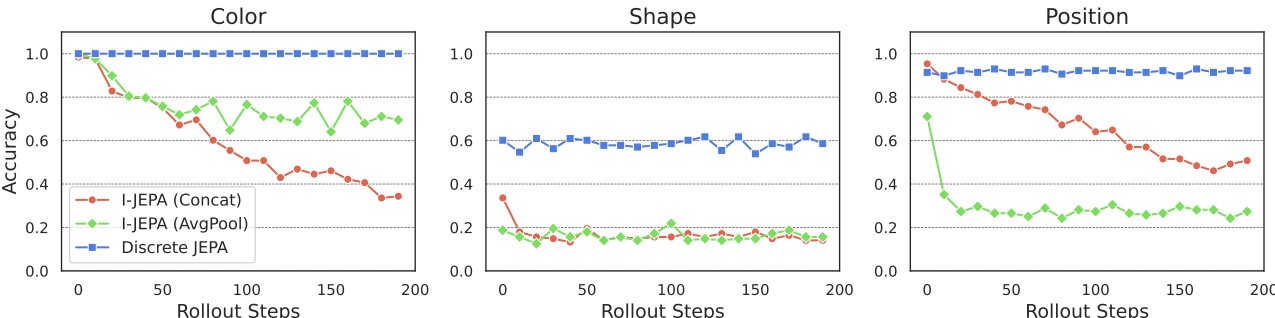

*Figure 3.* **Long-horizon prediction performance on Dancing-Sprites-Pattern dataset.** Performance comparison across color (left), shape (center), and position (right) prediction tasks over 200 rollout steps. Discrete-JEPA maintains stable performance while I-JEPA variants degrade over time due to accumulated errors in continuous space. D-JEPA achieves perfect color prediction stability, highlighting the benefits of discrete semantic tokenization for symbolic reasoning tasks.

original JEPA framework:

$$\mathcal{L}_{\text{P2P}} = \sum_{i \in \mathcal{M}} ||\bar{z}_p^{(i)} - g_\phi^{\text{P2P}}(z_p, i)||_2^2. \quad (10)$$

This objective ensures that our extension preserves the spatial prediction capabilities of the original JEPA framework.

**Unified Training Objective.** The complete training objective combines all predictive losses with the vector quantization commitment loss:

$$\mathcal{L}_{\text{total}} = \lambda_1 \mathcal{L}_{\text{S2P}} + \lambda_2 \mathcal{L}_{\text{P2S}} + \lambda_3 \mathcal{L}_{\text{P2P}} + \mathcal{L}_{\text{VQ}}. \quad (11)$$

where $\mathcal{L}_{\text{VQ}}$ includes the standard VQ commitment loss for the discrete semantic tokens. This unified predictive framework enables the learning of discrete semantic tokens that effectively capture global context, while continuous patch tokens provide detailed local information for complex reasoning tasks.

## 5. Experiments

**Datasets & Evaluation Protocol.** We evaluate Discrete-JEPA on two challenging visual sequence prediction tasks designed to assess symbolic reasoning and long-horizon planning capabilities. (1) *Dancing-Sprites-Pattern* consists of image sequences featuring a single object that follows various color transition patterns (`Linear`, `Repeat-2`, `Zigzag-3`, `Repeat-3`). Given 4 conditioning frames, we evaluate long-horizon prediction performance over approximately 200 time steps, measuring accuracy on color, shape, and position property classification tasks. (2) *Blinking-Ball* features sequences with four balls exhibiting interacting position and color patterns, requiring simultaneous tracking of spatial and chromatic dependencies. We assess prediction capabilities over approximately 1,000 rollout steps, measuring performance through pixel-wise reconstruction accuracy.

Both datasets provide controlled environments for evaluating symbolic reasoning capabilities while maintaining sufficient complexity to effectively distinguish between different tokenization approaches. Detailed dataset specifications and evaluation protocols are provided in Appendix A.

**Baselines.** We compare Discrete-JEPA against I-JEPA (Assran et al., 2023) as our primary baseline. I-JEPA represents the most direct comparison as it shares the same underlying architectural framework but operates with continuous representations rather than discrete tokens. For fair comparison, we adapt I-JEPA to the sequential prediction setting by training autoregressive world models on the continuous representations learned by I-JEPA. This baseline allows us to isolate the specific contribution of discrete semantic tokenization while controlling for architectural differences.

**Implementation Details.** Our implementation extends the I-JEPA framework with semantic tokenization and complementary prediction objectives. We train autoregressive world models using standard Vision Transformer architecture (Dosovitskiy et al., 2020) for long-horizon sequence prediction tasks. Complete implementation details, hyperparameters, and training configurations are provided in Appendix B.

### 5.1. Main Results

5.1.1. LONG-HORIZON SYMBOLIC PREDICTION TASKS

**Discrete Tokenization Mitigates Accumulated Prediction Errors.** A fundamental advantage of Discrete-JEPA emerges in its ability to prevent error accumulation over extended prediction horizons. By operating in a constrained discrete index space rather than continuous representations, Discrete-JEPA eliminates the compounding errors that plague continuous prediction approaches. This is demonstrated in Dancing-Sprites-Pattern color prediction, where Discrete-JEPA maintains perfect accuracy (1.0) across 200 timesteps while I-JEPA variants show substantial degrada-

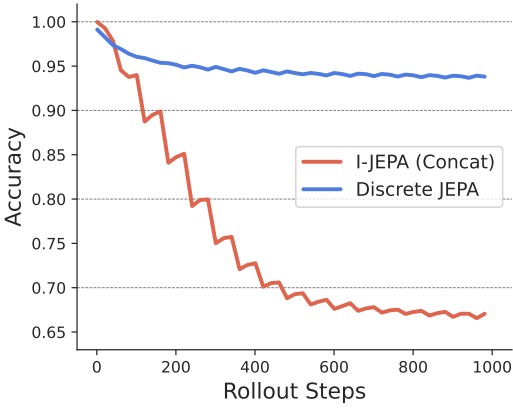

Figure 4. **Long-horizon prediction on Blinking-Ball task.** Discrete-JEPA maintains stable performance while I-JEPA degrades due to accumulated prediction errors, illustrating the benefits of discrete semantic tokenization for long-horizon sequence modeling.

tion (Figure 3), and in Blinking-Ball, where Discrete-JEPA stabilizes while I-JEPA exhibits continuous decline (Figure 4, Table 1).

**Semantic Abstraction Enables Robust Pattern Recognition.** Discrete-JEPA's semantic tokens, which integrate information across spatial patches, demonstrate superior capability for tasks requiring holistic understanding. This advantage is particularly evident in shape prediction tasks within Dancing-Sprites-Pattern, where semantic abstraction enables robust recognition of object-level properties. The approach effectively balances the need for high-level abstraction with sufficient detail retention for symbolic pattern modeling.

**Trade-off Between Abstraction and Spatial Precision.** While discrete semantic tokenization provides substantial benefits for symbolic reasoning, it involves a deliberate trade-off with fine-grained spatial information. This trade-off manifests in position prediction tasks, where I-JEPA (Concat) initially outperforms Discrete-JEPA due to explicit patch-level spatial encoding. However, the superior long-horizon stability of discrete approaches ultimately proves more valuable for extended sequence modeling. The multi-object complexity in Blinking-Ball further illustrates this trade-off, where Discrete-JEPA shows initial performance adjustment before achieving stable prediction, reflecting the increased demands of detailed positional reasoning in complex scenes.

### 5.1.2. VISUALIZATION OF PLANNING ON SEMANTIC SPACE

**Systematic Pattern Maintenance vs. Reactive Prediction.** Beyond quantitative performance metrics, Discrete-JEPA exhibits qualitatively distinct prediction behavior that

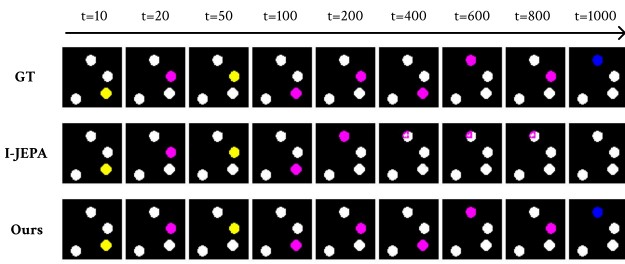

Figure 5. **Visualization of Semantic Planning on Blinking Ball.** Long-horizon predictions over 1,000 timesteps. I-JEPA breaks pattern consistency around t=600 despite initial accuracy, while Discrete-JEPA maintains systematic pattern integrity throughout, demonstrating deliberate planning in semantic token space. Additional visualization examples are provided in Appendix B.

suggests systematic planning capabilities rather than myopic next-step prediction. Figure 5 reveals this through extended sequence visualization on the Blinking-Ball task, where Discrete-JEPA maintains coherent pattern integrity throughout 1,000 timesteps while I-JEPA breaks systematic consistency around t=600 despite initially accurate predictions. This divergence indicates that Discrete-JEPA operates through deliberate pattern-based reasoning rather than reactive prediction.

**Evidence of Deliberate Reasoning in Semantic Token Space.** The preserved pattern consistency in Discrete-JEPA's predictions provides compelling evidence of Symbolic reasoning within the learned semantic token space. While I-JEPA's early accuracy suggests local prediction competence, its eventual pattern breakdown reveals the limitations of continuous representation for maintaining global symbolic consistency. In contrast, Discrete-JEPA's sustained adherence to underlying symbolic rules demonstrates that semantic tokenization enables the model to internalize and execute systematic reasoning processes, moving beyond immediate sensory-motor responses toward planned, rule-based behavior characteristic of deliberate cognitive processes.

## 6. Limitations and Future Work

Our approach presents several key limitations that open avenues for future research. (1) *Abstraction-Precision Trade-off*: Discrete semantic tokens excel at capturing high-level patterns but sacrifice fine-grained spatial information, evident in position prediction tasks where I-JEPA initially outperforms our method. (2) *Limited Scope*: Our evaluation focuses on controlled synthetic datasets that, while enabling precise assessment of symbolic reasoning, may not capture real-world complexity. (3) *Baseline Coverage*: Comparisons primarily involve I-JEPA, limiting our understanding

*Table 1.* **Blinking-Ball long-horizon prediction metrics.** I-JEPA shows better initial performance but continuous degradation, while Discrete-JEPA stabilizes after step 50 with superior long-horizon results (6× better LPIPS, 5× better MSE at 1000 steps).

| Metric | Method | Rollout Steps | | | | | | | |
|---|---|---|---|---|---|---|---|---|---|
| | | 10 | 20 | 50 | 100 | 200 | 400 | 800 | 1000 |
| LPIPS($\downarrow$) | Ours | 0.0028 | 0.0052 | **0.0099** | **0.0134** | **0.0189** | **0.0216** | **0.0245** | **0.0242** |
| | I-JEPA | **0.0001** | **0.0024** | 0.0114 | 0.0290 | 0.0578 | 0.1205 | 0.1538 | 0.1554 |
| MSE($\downarrow$) | Ours | 0.046 | 0.079 | 0.138 | **0.174** | **0.235** | **0.263** | **0.293** | **0.289** |
| ($\times 10^{-2}$) | I-JEPA | **0.003** | **0.031** | **0.131** | 0.337 | 0.654 | 1.263 | 1.449 | 1.461 |

relative to other contemporary tokenization approaches like VQGAN or TiTok.

Future work should address these limitations through several promising directions. (1) *Real-world Applications*: Evaluating Discrete-JEPA on robotics planning and complex video understanding tasks would validate its practical utility beyond controlled settings. (2) *Hierarchical Representation*: Developing multi-level semantic abstraction could address the abstraction-precision trade-off by maintaining tokens at different granularities.

Despite these limitations, our work establishes a promising foundation for advancing discrete semantic tokenization in latent predictive coding approaches, with demonstrated benefits for long-horizon prediction and compelling evidence of systematic reasoning capabilities.

## 7. Conclusion

This work addresses a key challenge in current visual representation learning: limitations of existing tokenization methods to effectively support symbolic reasoning and long-horizon planning. While recent advances in image tokenization have achieved remarkable success in reconstruction and generation tasks, they fail to capture the high-level semantic abstractions necessary for systematic inference and deliberate planning—capabilities central to cognitive intelligence.

We introduce Discrete-JEPA, which extends the Joint-Embedding Predictive Architecture with semantic-level discrete tokenization and complementary predictive objectives. Our approach learns discrete semantic tokens that capture global image context while preserving the benefits of latent-space predictive learning. Through carefully designed `S2P`, `P2S`, and `P2P` objectives, Discrete-JEPA enables effective information flow between semantic and spatial representations, resulting in robust tokenization for symbolic reasoning tasks.

Our experimental evaluation demonstrates clear advantages of discrete semantic tokenization over existing methods. On challenging visual sequence prediction tasks, Discrete-JEPA

significantly outperforms I-JEPA baselines, maintaining stable performance over extended horizons while continuous methods suffer from accumulated prediction errors. Most notably, our visualization analysis reveals compelling evidence of systematic pattern maintenance and deliberate reasoning behavior within the learned semantic token space—suggesting the emergence of semantic planning capabilities.

While our current evaluation focuses on controlled synthetic environments, the demonstrated capabilities suggest potential applications in robotics planning and multi-modal reasoning, though further evaluation on real-world tasks is needed. Discrete-JEPA represents a meaningful step toward developing AI systems that can perform deliberate reasoning rather than purely reactive prediction, establishing a foundation for future research in symbolic visual reasoning.

## Accessibility

We have embedded figures as text-readable PDF files wherever possible, including the use of vector graphics to support compatibility with screen readers and ensure clarity when zoomed. This manuscript has been prepared using the official ICML style files, which incorporate accessibility-focused formatting. Furthermore, we have provided explanatory text for all figures and tables in addition to their captions, to facilitate understanding for readers using assistive technologies.

## Acknowledgements

This work was supported by Institute of Information & Communications Technology Planning & Evaluation (IITP) grant funded by the Korea government (MSIT) (No. RS-2024-00509279, Global AI Frontier Lab). CH is supported by the DoD NDSEG Fellowship.

## Impact Statement

Our work advances AI systems with enhanced symbolic visual reasoning capabilities, offering significant benefits through improved systematic reasoning abilities. The discretization of latent visual representations enables a better understanding of AI visual processing, contributing to more trustworthy and coherent systems. These capabilities hold promise, particularly for scientific research, safer autonomous driving, and systematic logical reasoning tasks.

However, enhanced symbolic reasoning and planning abilities carry inherent risks requiring careful consideration. More capable AI agents could potentially be misused for surveillance, information manipulation, or autonomous decision-making without oversight. Improved planning capabilities might enable systems to pursue objectives in unexpected ways if misaligned. While our work represents early-stage research in highly synthetic and controlled settings, we encourage continued development of safety frameworks, noting that improved interpretability came with discretization, may facilitate better monitoring of future AI system behavior.

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

# A. Additional Details of Dataset Design

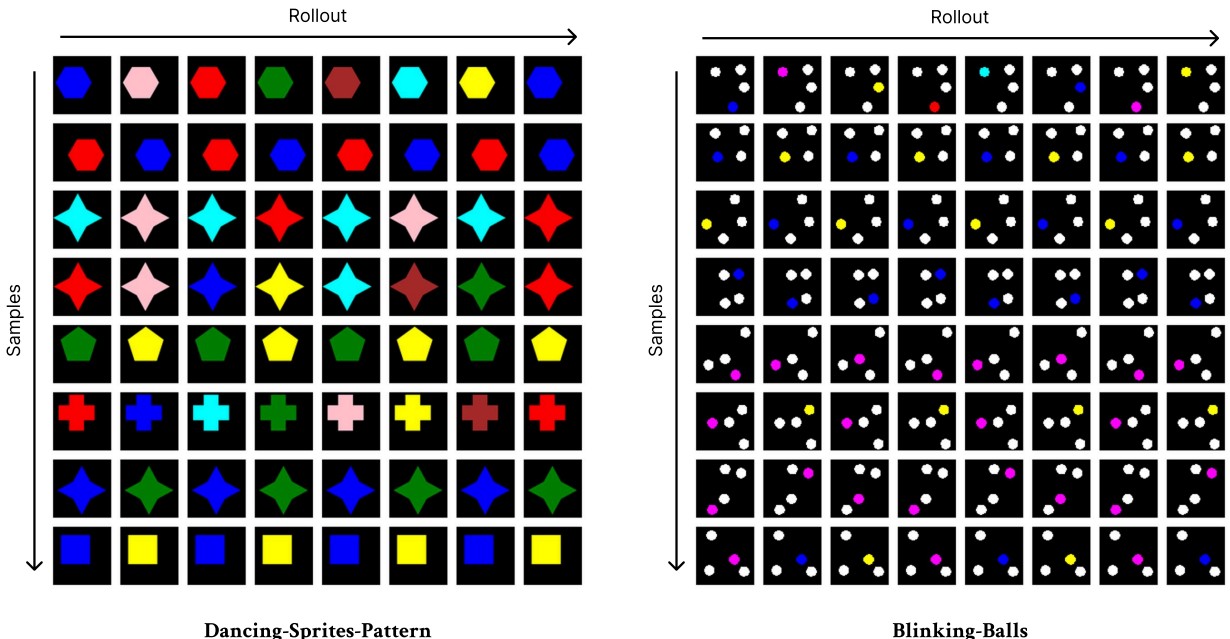

*Figure 6.* **Dataset Visualization.**

## A.1. Dancing-Sprites-Pattern Dataset

The Dancing-Sprites-Pattern dataset consists of 64×64 color image sequences, each containing a single object at a fixed spatial position. Built upon the Spriteworld environment (Watters et al., 2019), each object can take one of four possible shapes (star, square, circle, triangle) and one of seven possible colors. The dataset is designed to test the model's ability to learn and predict abstract color transition patterns over time.

**Pattern Types.** We define four distinct color pattern categories, where colors are indexed from 0 to 6:

1. `Linear`: Colors progress sequentially with fixed hop intervals (e.g., 2-hop: 0→2→4→6→1→3→5→0→...). Both starting color and hop size are randomly determined.

2. `Repeat-2`: Two randomly selected colors alternate repeatedly (e.g., 1→4→1→4→...).

3. `Zigzag-3`: Three randomly chosen colors follow a zigzag pattern (e.g., 1→5→7→5→1→5→7→...).

4. `Repeat-3`: Three randomly selected colors cycle sequentially (e.g., 1→5→7→1→5→7→...).

**Evaluation Protocol.** We employ a world model evaluation framework with a conditioning horizon of 4 frames and a prediction horizon of 4 frames. The world model takes 4 conditioning Discrete-JEPA tokens as input and predicts the subsequent 4 tokens. Predicted tokens are then processed through a pre-trained linear classifier to predict the color of each frame. Performance is measured using color prediction accuracy.

## A.2. Blinking-Ball Dataset

The Blinking-Ball dataset features 64×64 color image sequences containing four white ball objects at fixed spatial positions. Following the experimental protocol from Jiang et al. (2024), at each time step, exactly one ball is colored with one of five possible non-white colors, while the remaining three balls remain white. This dataset tests the model's capacity to simultaneously track positional and color patterns.

**Pattern Structure.** Each sequence follows two interconnected pattern types:

1. `Position Pattern`: A random permutation of the four balls determines the sequence in which balls will be colored across time steps.

2. `Color Pattern`: A randomly selected subset of the five available colors determines the color sequence applied to the balls according to the position pattern.

The interaction between position and color patterns creates complex temporal dependencies that require both spatial and chromatic reasoning capabilities.

**Evaluation Protocol.** We use a conditioning horizon of 6 frames and evaluate prediction performance over the subsequent 6 frames. Predicted Discrete-JEPA tokens are decoded into images using a pre-trained image decoder, and performance is assessed through pixel-wise classification accuracy of the reconstructed sequences. Both datasets provide controlled environments for evaluating symbolic reasoning capabilities while maintaining sufficient complexity to distinguish between different tokenization approaches. The fixed spatial layouts allow models to focus on learning temporal color and position patterns without the confounding factor of spatial prediction.

# B. Additional Implementation Details

## B.1. World Model Architecture

Our world model employs a Vision Transformer (Dosovitskiy et al., 2020) architecture to perform autoregressive prediction over token sequences. Given $H_c$ conditioning tokens from previous timesteps, the model predicts tokens for the subsequent $H_p$ prediction timesteps. The model then repeats this procedure autoregressively to predict additional future frames over extended horizons.

We implement different world model variants tailored to each baseline's representation type:

### B.1.1. DISCRETE-JEPA WORLD MODELS

**R2I (Representation-to-Index)**: Takes quantized tokenizer output vectors as input and predicts tokenizer output indices for future timesteps using CrossEntropy loss.

**I2I (Index-to-Index)**: Takes quantized tokenizer output indices, re-embeds them through learned embeddings, and predicts tokenizer output indices for future timesteps using CrossEntropy loss.

### B.1.2. I-JEPA WORLD MODELS

**R2R-Concat (Representation-to-Representation, Concatenated)**: Uses all 64 I-JEPA patch tokens as input to predict future patch token vectors using MSE loss.

**R2R-AvgPool (Representation-to-Representation, Average Pooled)**: Uses averaged I-JEPA patch tokens (single pooled token) as input to predict future averaged patch token vectors using MSE loss.

## B.2. Image Decoder for Blinking-Ball Task

For the Blinking-Ball task, we implement a Transformer decoder without causal masking to enable pixel-level reconstruction from learned token representations. The decoder follows a standard Transformer architecture where tokenizer outputs serve as key and value vectors in the cross-attention layers.

**Input preprocessing** is carefully designed to ensure the tokenizer focuses exclusively on ball identification and coloring. We preprocess input images by resetting all ball colors to white (the default state) before patchifying and feeding them to the transformer. This design choice forces the model to rely on the tokenizer output for determining which balls should be colored and with what colors.

**Output prediction** is formulated as a pixel-wise classification problem over 7 possible classes: black background, white balls, and 5 possible ball colors. The model is trained using CrossEntropy loss to predict the correct color for each pixel location.

*Table 2.* Hyperparameters for Dancing-Sprites-Pattern Experiments

| Component | Hyperparameter | Model | |
|---|---|---|---|
| | | Discrete-JEPA | I-JEPA |
| Encoder Configuration | Architecture | ViT-Base | ViT-Base |
| | Patch Size | 8 | 8 |
| | Random Masking | 40%-60% | 40%-60% |
| | Learning Rate | 1e-5 | 1e-5 |
| | LR Schedule | 5% warmup + cosine | 5% warmup + cosine |
| | Batch Size | 128 | 128 |
| Tokenization | Quantizer | SVQ | - |
| | Semantic Tokens | 8 per image | - |
| | Token Dimension | 96 | 768 |
| | Codebook Size | 1024 | - |
| | VQ Learning Rate | 1e-5 (15% warmup) | - |
| World Model | Architecture | Transformer Encoder | Transformer Encoder |
| | Layers | 2 | 2 |
| | Attention Heads | 4 | 4 |
| | Hidden Dimension | 96 | 768 |
| | Input Tokens | 8 per image | 64 (Concat) / 1 (Pool) |
| Property Prober | Architecture | AvgPool + Linear | AvgPool + Linear |
| | Optimizer | LARS | LARS |
| | Learning Rate | 0.1 | 0.1 |
| | Training Steps | 20,000 | 20,000 |

### B.3. Hyperparameters

We provide comprehensive hyperparameter configurations for both experimental tasks to ensure reproducibility and fair comparison between Discrete-JEPA and I-JEPA baselines. Dancing-Sprites-Pattern uses smaller models due to simpler visual patterns, while Blinking-Ball requires larger capacity for complex spatial-temporal dependencies. Tables 2 and 3 detail the configurations for each dataset.

## C. Additional Visualization Results for Semantic Planning

This section provides additional visualization examples that complement the semantic planning analysis in Figure 5 of the main paper. Figure B shows three additional Blinking Ball sequence instances, demonstrating consistent pattern maintenance behavior across different initial conditions. These examples reinforce our findings that Discrete-JEPA maintains systematic pattern integrity while I-JEPA exhibits pattern breakdown during long-horizon prediction, providing robust evidence for emergent planning capabilities in semantic token space.

*Table 3.* Hyperparameters for Blinking-Ball Experiments

| | | Model | |
|---|---|---|---|
| Component | Hyperparameter | Discrete-JEPA | I-JEPA |
| Encoder Configuration | Architecture | ViT-Base | ViT-Base |
| | Patch Size | 8 | 8 |
| | Random Masking | 50%-70% | 50%-70% |
| | Learning Rate | 1e-5 | 1e-3 |
| | Optimizer | AdamW | AdamW |
| | LR Schedule | 5% warmup + cosine | 5% warmup + cosine |
| | Batch Size | 128 | 128 |
| | Training Steps | 300K | 300K |
| Tokenization | Quantizer | SVQ | - |
| | Semantic Tokens | 32 per image | - |
| | Token Dimension | 96 | 768 |
| | Codebook Size | 1024 | - |
| | VQ Learning Rate | 1e-5 (15% warmup) | - |
| World Model | Type | I2I (Index-to-Index) | R2R (Repr-to-Repr) |
| | Architecture | Transformer Encoder | Transformer Encoder |
| | Layers | 2 | 2 |
| | Attention Heads | 4 | 4 |
| | Hidden Dimension | 768 | 768 |
| | Input Tokens | 32 per image | 32 per image |
| | Learning Rate | 1e-3 | 1e-3 |
| | Optimizer | AdamW | AdamW |
| | LR Schedule | 5% warmup + cosine | 5% warmup + cosine |
| | Training Steps | 15K | 15K |
| | Embedding | Index $\rightarrow$ 768dim | - |
| Decoder | Architecture | Transformer w/o causal mask | Transformer w/o causal mask |
| | Layers | 3 | 3 |
| | Attention Heads | 4 | 4 |
| | Hidden Dimension | 64 | 64 |
| | Input Projection | - | 768 $\rightarrow$ 64 linear |
| | Learning Rate | 1e-3 | 1e-3 |
| | Optimizer | AdamW | AdamW |
| | LR Schedule | 5% warmup + cosine | 5% warmup + cosine |
| | Training Steps | 50K | 50K |

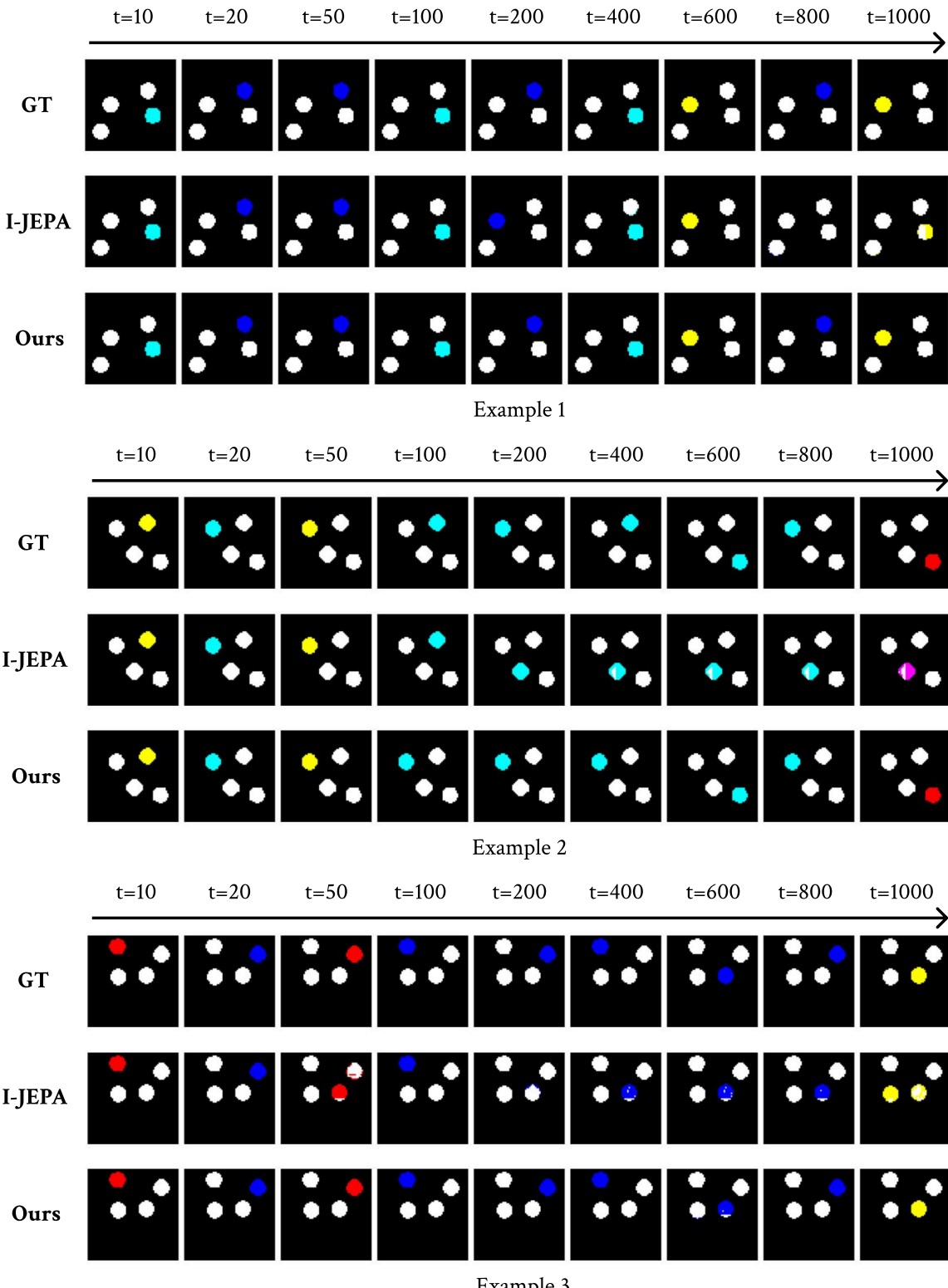

*Figure 7.* Additional Visualization of Semantic Planning on Blinking Ball.

