# OpenReview forum: "Discrete JEPA: Learning Discrete Token Representations without Reconstruction"
_ICML.cc/2025/Workshop/TokShop — TokShop_

### Official Review · Reviewer_ThE9 · 2025-06-06
**Discrete-JEPA is an extension of the JEPA framework that works with discrete, quantized global and continuous patch-level representations. Both features are used in synthetically generated tasks, proving significant improvements.**

**Rating:** 8
**Confidence:** 5

**Review:**

Summary:
Discrete-JEPA is an extension of the JEPA framework that works with both discrete, quantized global and continuous patch-level representations. The embeddings are learned in a self-supervised distillation cross-scenario. Both features are further used in auto-regressive downstream classification tasks. The tasks are syntentically generated and operate with simple features such as geometric shape, color, position, and do not deal with advanced object detection tasks. However, Discrete JEPA achieves better and consistent performance than I-JEPA.

Strengths:
 - Discrete-JEPA is a good extension of the JEPA framework, providing consistent performance in synthetic evaluation datasets, significantly better than I-JEPA (Concat and AvgPool).
 - The self-distillation learning scenario allows the model to learn not only local features from global discrete representations but also global and local features from masked patch representations.

Weakness:
- The Figure 2 caption could be more explainable in plain words, even if the formulae from section 4.1 are more precise.  e.g. The images are encoded with semantic discrete ($z_s$ or blue) and patch-level continous ($z_p$ or green) representations. Also, I would include a small visual representation of the features that are used in the downstream tasks (in all scenarios, like Representation-to-Index and Index-to-Index)

Typos and revisions:
- The formulae from sections 3 and 4 would look much better numbered, and you can easily refer to them after.
- Figure 2, the "Encoder" would look clearer as "Context Encoder". Also, the "L2_loss" might look better under the bar, and to be consistent, either "l2_loss" or "L2_Loss".
- $z^{\_}_{p}$ "from" (Line 216) is a bit misleading, maybe it is better to be removed, or the be written inside text, or to refer to the previous formula, from Lines 170-171.
- Are the synthetic datasets for the downstream tasks completely generated by yourself? If so, this should be more clearly mentioned in the experiments section. Otherwise, the source should be properly cited.
- For lambda 1-3 and VQ, which values were used as hyperparameters? Maybe it's better to be mentioned in Table 2

---

### Official Review · Reviewer_gpJK · 2025-06-07
**Accept - Novel and interesting token representation scheme**

**Rating:** 7
**Confidence:** 3

**Review:**

Strengths:
* The proposed Discrete-JEPA framework is a novel idea worth considering that extends existing JEPA.
* Discrete JEPA learns to encode global semantic information into discrete tokens while preserving fine-grained spatial details
 through complementary continuous representations. This is an important achievement of D-JEPA.
* Through experimental results manuscript shows that "Discrete Tokenization Mitigates Accumulated Prediction Errors. ", "Semantic Abstraction Enables Robust Pattern Recognition.".
* Over two separate datasets D-JEPA shows significant accuracy improvements over I-JEPA.

Weaknesses:
* The scope of evaluation is limited with only datasets being considered. A more real world and larger dataset would have been more helpful.
* Limited baseline. While comparing to I-JEPA makes sense, it may not be the best baseline here given that D-JEPA is more focused on global semantic information.

Some parts in the conclusion are hard to justify based on the paper. For example:

> While our current evaluation focuses on controlled synthetic environments, the demonstrated capabilities suggest
promising applications in robotics planning, multi-modal reasoning, and other domains requiring systematic inference

I am not sure this conclusion follows from whatever that is presented in the paper.

> Our experimental evaluation demonstrates clear advantages of discrete semantic tokenization over existing methods.

Existing method here being just I-JEPA and not entire universe of existing methods.

---

### Official Review · Reviewer_hPGq · 2025-06-08

**Rating:** 7
**Confidence:** 2

**Review:**

This paper presents DiscreteJEPA a new tokenization method, or token representation method, that is combining 2 methods for tokenization in computer vision. DiscreteJEPA combines the discreteness of tokens, which is needed for contemporary SOTA CV models, while also uses autoencoding to learn representations in a continuous space that are aware of the semantics of the image. The new method then outputs tokens that are both semantics-aware and discrete.

The authors demonstrate the superiority of their method by comparing its performance to that of I-JEPA (continuous semantically-aware method) on 2 visual sequence prediction tasks.

All in all, the work in this paper seems to be decent and fill in a specific gap in a way that exploits the advantages of both existing approaches. The main weakness that I could spot in this work is the lack of comparison to other discrete tokenization methods that DiscreteJEPA aspires to replace. The authors mention TiTok as a contemporary SOTA discrete tokenization, but the experiments do not include any comparison to it.

(I must disclose here, that I may have missed many strengths and weaknesses of this paper as I am not a CV person and I lack a lot of knowledge that is trivially mentioned by the authors)

---

### Decision · Program_Chairs · 2025-06-10

Accept